# The Therapeutic Potential of Exosomes from Mesenchymal Stem Cells in Multiple Sclerosis

**DOI:** 10.3390/ijms251910292

**Published:** 2024-09-24

**Authors:** Torbjørn Kråkenes, Casper Eugen Sandvik, Marie Ytterdal, Sonia Gavasso, Elisabeth Claire Evjenth, Lars Bø, Christopher Elnan Kvistad

**Affiliations:** 1Neuro-SysMed, Department of Neurology, Haukeland University Hospital, 5021 Bergen, Norwaylars.bo@helse-bergen.no (L.B.); christopher.kvistad@helse-bergen.no (C.E.K.); 2Department of Clinical Medicine, Faculty of Medicine, University of Bergen, 5021 Bergen, Norway

**Keywords:** exosomes, mesenchymal stem cells, microglia, multiple sclerosis, neuroprotection, neuroregeneration, remyelination

## Abstract

Although treatment for multiple sclerosis (MS) has undergone a revolution in the last decades, at least two important barriers remain: alleviation of innate inflammation driving disease progression and promotion of remyelination and neural regeneration. Mesenchymal stem cells (MSCs) possess immunomodulatory properties and promote remyelination in murine MS models. The main therapeutic mechanism has, however, been attributed to their potent paracrine capacity, and not to in vivo tissue implantation. Studies have demonstrated that exosomes released as part of the cells’ secretome effectively encapsulate the beneficial properties of MSCs. These membrane-enclosed nanoparticles contain a variety of proteins and nucleic acids and serve as mediators of intercellular communication. In vitro studies have demonstrated that exosomes from MSCs modulate activated microglia from an inflammatory to an anti-inflammatory phenotype and thereby dampen the innate inflammation. Rodent studies have also demonstrated potent immunomodulation and remyelination with improved outcomes following exosome administration. Thus, exosomes from MSCs may represent a potential cell-free treatment modality to prevent disease progression and promote remyelination in MS. In this narrative review, we summarize the current knowledge of exosomes from MSCs as a potential treatment for MS and discuss the remaining issues before successful translation into clinical trials.

## 1. Multiple Sclerosis

Multiple sclerosis (MS) is an immune-mediated disease of the central nervous system (CNS) characterized by inflammation-causing multifocal demyelination and subsequent neuronal degeneration. MS represents the most common non-traumatic cause of disability in young European adults and affects approximately 2.8 million people worldwide, with an increasing incidence over the past decades [1]. All parts of the CNS may be affected by MS, and common symptoms are motor and sensory deficits, fatigue, pain, visual impairment and cognitive dysfunction.

The core MS phenotypes are those of relapsing-remitting disease (RRMS) dominated by inflammation mediated by the adaptive immune system, and progressive disease, dominated by chronic microglia-driven inflammation and neuronal degeneration. RRMS is characterized by the invasion of macrophages, microglia and lymphocytes through the injured blood–brain barrier (BBB) into the CNS parenchyma with the formation of inflammatory and demyelinating lesions in both white and grey matter. Microglial activation has been found in the white matter remote from the active lesions, where they may represent the earliest stage of lesion development [2]. Microglia also play a role in the clearance of myelin debris, which is necessary for remyelination to occur. The remyelination in patients is, however, incomplete with thinner and more fragile myelin sheaths than normal myelin [3].

After typically 15–20 years, patients may experience a progressive clinical course termed secondary progressive MS (SPMS), whereas approximately 15% of patients have a progressive course from the beginning, termed primary progressive MS (PPMS). Neurodegeneration is the hallmark of progressive MS disease, which typically leads to immobility and cognitive decline. Progressive MS is also characterized by a compartmentalized chronic inflammation with focal and diffuse microglial activation behind an intact BBB.

There is no curative treatment available for MS. Current treatment recommendations for RRMS and progressive forms of MS include the use of CD20 antibodies, such as rituximab, ocrelizumab, or ofatumumab, which have demonstrated high efficacy in preventing MS inflammatory disease activity [4]. Hematopoietic stem cells have also demonstrated good results with approximately 70% showing no evidence of new disease activity after 5 years [5]. There is, however, no treatment available for the chronic compartmentalized inflammation within the CNS, responsible for neural degeneration and disease progression. This persistent inflammation is primarily caused by microglial activation, both in the normal-appearing white matter (NAWM) and in the rim of chronic active, so-called “smoldering”, lesions [6].

Smoldering lesions are present in 56% of the general MS population, and represent a direct cause of progression to disability in both RRMS and progressive MS [7]. In RRMS, this becomes apparent as up to 30% of patients experience accumulation of disability despite immunomodulatory treatment, of which a majority is being caused by progression independent of relapse activity (PIRA) [8]. In progressive MS, smoldering lesions are evident in up to 78% of patients, and have been associated with a more aggressive disease course [9]. The chronically activated microglia, which are sealed off behind the BBB, have proven to be difficult to target with conventional pharmacological agents. Consequently, there is no effective therapy for this type of innate inflammation.

Another barrier in modern MS treatment is the lack of means to promote neuroregeneration. Despite some degree of spontaneous remyelination in the first years of MS disease, the newly formed myelin is thin and unable to metabolically support axons over time [10]. This leads to neural degeneration and progressive disease. Currently, there are no approved therapies for remyelination or axonal regeneration. Once neuronal degeneration is present, the progression and resulting disability is irreversible. Consequently, the burden of MS remains substantial, both from an individual point of view as well as from a socioeconomic perspective. In the US alone, costs related to labor productivity loss, caregiving and home modification due to MS were estimated to $22.1 billion in 2019 [11]. There is thus an urgent need for new, innovative therapies to address the current challenges in MS treatment.

## 2. Mesenchymal Stem Cells and Their Secretome

Mesenchymal stem cells (MSCs) are characterized by their self-renewal potential and multipotent properties. MSCs do not have a unique cell marker but are defined by the presence and absence of different cluster of differentiation (CD) markers and tri-lineage differentiation potential in vitro [12]. MSCs are present in most tissues and have been isolated from multiple adult tissue types, including bone marrow (BM), fat and dental pulp. Within the BM, MSCs primarily function to help maintain the hematopoietic stem cell developmental niche. It is, however, clear that they also play a significant systemic role in immunomodulation and repair.

Due to their accessibility, the use of autologous or allogeneic MSCs does not represent any ethical concerns, as may be the case with neural, embryonal or fetal stem cells. Furthermore, unlike induced pluripotent stem cells, genetic manipulation is not necessary. This, combined with their intrinsic regenerative capacity, has made MSCs attractive candidates for testing in human trials. MSCs have also been demonstrated to migrate towards chemokines released after CNS injury, thus increasing their therapeutic value [13,14].

Several studies have shown that MSCs mediate anti-inflammatory effects and promote remyelination in different murine MS models, resulting in improved outcomes [15,16,17]. This was recently summarized in a meta-analysis, including 88 studies applying different MS models [18]. Clinical studies in MS patients have also demonstrated safety and indications of immunomodulation with beneficial outcomes in RRMS and active progressive MS [19,20,21]. Since intravenous application of MSCs has not proven to be effective [22], presumably due to entrapment of MSCs in the lungs [23], intrathecal administration has been proposed as the preferred route of administration [24]. However, there is little evidence concerning the fate of MSCs, once intrathecally injected into patients. Drawbacks of MSC treatment also include high production costs to reach sufficient cell numbers, heterogeneity within the cell population, and cell-related complications due to the invasive form of treatment, such as arachnoiditis [25].

Administration of living cells may, however, not be necessary to harness the beneficial therapeutic effects of MSCs. Their therapeutic function has mainly been related to their potent paracrine capacity, and not to in vivo tissue implantation and trans-differentiation [26,27]. MSCs are highly secretory, and their secretome contains a soluble and a vesicular fraction (Figure 1). The soluble fraction contains a plethora of neurotrophic growth factors, chemokines and cytokines such as IL-6, IL-10, IL-17, PGE2, CXCL-10, glial-derived neurotrophic factor (GDNF), brain-derived neurotrophic factor (BDNF), vascular endothelial growth factor (VEGF), fibroblast growth factor (FGF), hepatocyte growth factor (HGF), nerve growth factor (NGF) and insulin-like growth factors 1 and 2 (IGF-1 and IGF-2) [28,29]. The vesicular fraction contains extracellular vesicles (EVs) of different sizes, including exosomes.

Studies have demonstrated that the secretome of MSCs reduces inflammation through inhibition of T cells and modulation of macrophages resulting in decreased production of pro-inflammatory cytokines and improved outcomes in murine MS models [30,31]. The MSC secretome has also been shown to enhance remyelination by promoting oligodendrocyte differentiation, resulting in functional improvement of mice induced with experimental autoimmune encephalomyelitis (EAE) [32]. Emerging research indicates that EVs are essential contributors to the therapeutic effects of MSCs and their secretome.

## 3. Exosomes

Exosomes are a subpopulation of EVs produced and secreted by virtually all cells. Exosomes are characterized by their spherical morphology, lipid bilayer membrane, and the presence of specific membrane proteins such as the tetraspanins CD9, CD63, CD81, and CD82. These vesicles range from 30 to 150 nm in diameter and contain nucleic acids (DNA and RNA), lipids, proteins, and metabolites [33]. The lipid membrane protects the cargo from degradation. Exosomes function as mediators of cell communication and regulate the fate of recipient cells by inducing changes in gene expression and cellular function [34].

EVs are typically divided into three subgroups based on their diameter: exosomes (30–150 nm), microvesicles (100–1000 nm), and apoptotic bodies (≥1 μm) [33,35]. Unlike microvesicles and apoptotic bodies, which are products of plasma membrane shedding, exosomes are released from cells through the endocytic pathway. Briefly, multivesicular bodies (MVBs) are formed which eventually fuse with the plasma membrane to release the exosomes. Exosomes have a disc-like structure and flat spherical shape when viewed under an electron microscope (Figure 2).

More than 300 proteins [36] and 150 microRNAs (miRNAs) [37] have been identified inside exosomes of MSCs in addition to other biomolecules. miRNAs are noncoding RNAs regulating the expression of protein-coding genes. Once delivered, exosomes may regulate the fate and morphology of recipient cells by modulation of signaling pathways through the different miRNAs.

In many ways, exosomes can be regarded as miniature versions of the originating cell type. For stem cells, exosomes may, therefore, exhibit therapeutic effects similar to those of their parental cells. In contrast to living cells, exosomes are not self-replicating, have low immunogenicity, and may cross the BBB due to their size, polarity, and surface protein expression. Thus, exosomes may be employed as cell-free therapeutic vehicles owing to their unique abilities.

### 3.1. Exosome Biosynthesis

The biosynthesis of exosomes is initiated with endosomal maturation, involving distinct modifications of the endosomal membrane [38]. During this process, intraluminal vesicles (ILVs) are generated by the process of invagination, resulting in the development of multivesicular bodies (MVBs). These MVBs have the potential to be transported either to lysosomes for degradation, or to the plasma membrane for exocytosis, thereby releasing ILVs as exosomes into the extracellular environment. The specific mechanisms that determine whether exosomes escape degradation are not yet fully understood. Exosome biosynthesis involves three main stages: (1) MVB formation, (2) cargo sorting, and (3) MVB transport and fusion with the plasma membrane [38,39].

#### 3.1.1. MVB Formation

There are currently two known main pathways for exosome biogenesis: One being dependent on the endosomal sorting complex required for transport (ESCRT) and one being independent of this mechanism.

(1) ESCRT-dependent pathway: In this synthesis pathway, the ESCRT machinery is central for the formation of intraluminal vesicles (ILVs) within the multivesicular bodies (MVBs). This process involves certain vacuolar protein sorting (VPS) proteins along with other specific proteins that assemble into four essential complexes: ESCRT-0, -I, -II, and -III, along with accessory proteins such as ALIX (programmed cell death 6-interacting protein), TSG101 (tumor susceptibility gene 101 protein), and VPS4 [38]. ESCRT-0 initiates this process by binding to PI3P (phosphatidylinositol-3-phosphate), which captures ubiquitinated cargo. ESCRT-I, in cooperation with TSG101, recruits ESCRT-II to form a complex that facilitates further cargo capture. Subsequently, ESCRT-III forms spiral oligomers, which facilitate membrane constriction and inward budding. The final step involves ATP hydrolysis by VPS4 to disassemble the ESCRT complex, thereby allowing for subunit turnover. After this process, some proteins remain ubiquitinated, even after ILV synthesis. The ESCRT machinery interacts with protein tyrosine phosphatase (HD-PTP) to deubiquitinate these proteins, which is important for exosome function [40].

(2) ESCRT-independent pathways: ILVs may also be generated through ESCRT-independent mechanisms. One such pathway includes the nSMase2-ceramide pathway, which facilitates the formation of MVBs by inducing the organization of lipid raft microdomains [41]. Tetraspanins such as CD9, CD63, CD81, and CD82, which are abundant in exosomes [42], may regulate ILV formation through their interactions within these microdomains [43]. ESCRT-independent exosome biogenesis and secretion also involve the activation of certain Rab GTPases, particularly RAB31 [44]. After phosphorylation by the epidermal growth factor receptor (EGFR), RAB31 becomes activated and associates with flotillin proteins in lipid raft microdomains. This active form of RAB31 is essential for directing the EGFR into ILVs within MVBs without the involvement of the ESCRT machinery. Furthermore, RAB31 recruits the GTPase-deactivating protein TBC1D2B, which further inactivates RAB7. This action prevents MVB-lysosome fusion, thereby allowing MVBs to avoid degradation and thus facilitating the secretion of ILVs as exosomes.

#### 3.1.2. Cargo Sorting

Exosomal cargo includes a variety of biomolecules, which include nucleic acids, such as DNA, mRNA, miRNA, lncRNA, and circRNA. The contents of the MVBs also are altered through the trans-golgi «sorting» network (TGN). The RNA content of exosomes is unique and reflects the properties of the parental cell, with specific miRNAs being selectively packaged. miRNAs undergo a complex maturation process and possess the ability to either completely stop translation or promote mRNA degradation in recipient cells [45].

The sorting of miRNAs into exosomes involves interactions with RNA-binding proteins and other proteins involved in exosome biogenesis, such as ALIX, nSMase2 (neural sphingomyelinase 2), and VPS4 [46]. Specific motifs within miRNAs facilitate their loading into exosomes, suggesting a regulatory sorting process. In addition to nucleic acids and tetraspanins, exosomes also carry various other proteins, including heat shock proteins (HSP70 and HSP90), MVB formation-related proteins, and other cytosolic and membrane proteins.

Lipids are another significant component of exosome cargo that contribute to membrane structure and function. Exosomes contain high concentrations of cholesterol, sphingomyelin and ceramide, all of which are involved in membrane fluidity and curvature. Exosomes also carry metabolites that can influence the metabolic state of recipient cells [47]. Metabolite content includes amino acids, lipids, sugars, and organic acids, which can modulate various biochemical pathways [48]. RNA-binding proteins such as Ago2, heterogeneous nuclear ribonucleoprotein A2/B1 (hnRNPA2B1), and SYNCRIP play a critical role in RNA sorting into exosomes. SYNCRIP binds specific motifs in the 3′-UTR of miRNAs, facilitating their inclusion in exosomes. Other RNA-binding proteins, such as YBX1, have been implicated in the selective packaging of miRNAs into exosomes [46].

#### 3.1.3. MVB Transport, Fusion with the Plasma Membrane and Exocytosis

The ESCRT-dependent and independent pathways for MVB transport and exocytosis are not mutually exclusive and can occur concurrently, resulting in a heterogeneous population of ILVs within the same MVB [38]. The biogenesis of MVBs and cargo sorting overlap and happen simultaneously. After cargo sorting in MVBs, the vesicles merge with the plasma membrane, and release their contents as exosomes (Figure 3). Regulatory signals, such as the activity of RAB31 and recruitment of TBC1D2B, determine whether MVBs fuse with lysosomes or the plasma membrane by inhibiting RAB7 [44]. Exosomes tagged for extracellular release are transported to the plasma membrane with the help of the actin and microtubule cytoskeletons. RAB27A/B tether the MVBs to the kinesins for this transportation. The docking to the plasma membrane is facilitated by RAB GTPases such as RAB27A/B, RAB11 and RAB35, and fusion is facilitated by SNARE proteins, allowing exosomes to enter the extracellular space and interact with target cells [49].

### 3.2. Interaction with Recipient Cell

Upon release, exosomes move across the surface of target cells until they reach suitable microdomains for internalization and endocytosis [40]. Generally, two mechanisms are involved in the interaction between exosomes and target cells. (1) Exosomes enter the target cells through micropinocytosis or endocytosis, or (2) they fuse with the cellular membrane and release their cargo to activate various signaling pathways [50]. The latter appears to be elicited by the binding of exosome membrane proteins to their respective cell-surface molecules. The proteins that are responsible for these interactions in EVs are tetraspanins, integrins, proteoglycans, and lectins. Membrane fusion, which is governed by syncytins, starts with their high-affinity binding to the trans-membrane proteins MFSD2a or ASCT2 on the target cells [51]. The fusion process is completed when syncytins insert their hydrophobic sequences into the plasma membrane of the cell, causing lipids to reorganize and proteins to restructure, followed by membrane dimpling. Alternatively, exosomes interact with the recipient cells through ligand-receptor binding on the surface to stimulate cascade responses. In some cases, the binding itself can induce internal signaling; however, in most cases, signaling is induced after membrane fusion, endocytosis, and/or the discharge of luminal cargo into the cytosol of the recipient cell [33].

### 3.3. Effects of MSC Exosomes in the CNS

Microglia are considered the primary immune cells in the CNS and play a vital role in MS pathogenesis by contributing to both neuroinflammation and neurodegeneration [52]. In MS, microglia respond to CNS injury and inflammation by adopting a pro-inflammatory M1 phenotype, secreting cytokines such as TNF-α, IL-1β, and IL-6. These cytokines promote demyelination and neuronal damage. The prolonged activation of microglia in this state exacerbates neuroinflammation and contributes to chronic neurodegeneration and lesions observed in MS [53,54]. Although the M1/M2 polarization concept is considered to be oversimplified, it provides a useful framework for understanding the various roles played by microglia in neuroinflammatory conditions.

Exosomes derived from MSCs appear to have promising therapeutic agents to modulate microglial activity. Studies have shown that MSC exosomes can induce a shift in microglial polarization from the pro-inflammatory M1 phenotype, toward an anti-inflammatory M2 state. This shift is characterized by a decrease in the expression of pro-inflammatory cytokines and an increased expression of anti-inflammatory cytokines such as IL-10 and TGF-β [55,56]. Additionally, the phagocytic activity of microglia is essential, supporting the clearance of apoptotic cells and myelin debris, the latter of which inhibits remyelination. This removal is essential for remyelination and neuronal survival in MS.

Specific miRNAs carried by MSC exosomes play a significant role in the regulation of microglial polarization. The overexpression of miR-216a-5p in exosomes derived from hypoxic bone marrow MSCs (BM-MSCs) has been shown to reverse the secretion of inflammatory factors by microglia, including TNF-α, IL-6, and iNOS (inducible nitric oxide synthase) [55]. Moreover, miR-146a-5p [57] and miR-125a [58] have also been demonstrated to alleviate pro-inflammatory microglia following CNS injury. Exosomes from MSCs modulate microglia from inflammatory to anti-inflammatory phenotype by inhibition of the TLR4/NF-κB/PI3K/AKT inflammatory cascade [55]. This was demonstrated in a rodent model where exosomes were administrated intravenously as a single injection after SCI. This led to significantly improved outcomes in mice receiving exosomes as compared to controls. Interestingly, exposure of MSCs to hypoxia improved the anti-inflammatory effect of the resulting exosomes.

These miRNAs promote the transformation and conversion of microglia from a pro-inflammatory state to an anti-inflammatory state by increasing the population of CD206^+^ microglia and reducing CD16/32^+^ microglia. CD206^+^ microglia are associated with anti-inflammatory effects. In contrast, CD16/32^+^ microglia are considered inflammatory due to their role in enhancing the production of inflammatory mediators and cytokines. The shift towards CD206^+^ microglia is, therefore, associated with reduced inflammation and a better anti-inflammatory response. This transformation leads to the downregulation of CysLT2R (cysteinyl leukotriene receptor 2), which is partially responsible for the inflammatory response of microglia.

The mechanisms through which MSC exosomes regulate microglia include signaling pathways such as the TLR4/NF-κB/PI3K/AKT cascade, which is known to influence the polarization of microglial phenotypes [55]. This pathway is influenced by the miRNA miR-216a-5p, which has the ability to downregulate the expression of TLR4, thereby leveraging microglia towards an M2 anti-inflammatory direction.

Exosomes applied for therapeutic purposes also impact oligodendrocytes and neurons within the CNS. Previous studies have demonstrated that exosomes from stem cells are colocalized with neurons and oligodendrocytes following intranasal administration, suggesting that the exosomes are absorbed by these cells [59]. Another study applying the cuprizone and EAE models in mice, demonstrated an increased number of newly generated oligodendrocytes in addition to mature oligodendrocytes with higher levels of myelin basic protein following intravenous administration of exosomes from MSCs [60]. Likewise, an in vitro model of ischemic stroke showed a beneficial effect of MSC exosomes as miR-134 promoted survival in oligodendrocytes by negatively regulating the caspase 8-dependent apoptosis pathway. There is also some evidence that exosomes may affect neurons. In a rodent model of Alzheimer’s disease, MSC-derived exosomes inserted via stereotaxic surgery resulted in an increased number of neural precursor cells in the subventricular zone of the brain, suggesting that exosomes accelerated neurogenesis [61]. Findings were associated with improved cognitive results in mice receiving exosomes. Exosomes have also been shown to promote neurogenesis and improve outcomes in models of ischemia [62] and traumatic brain injury [63].

Taken together, evidence from pre-clinical studies indicates that exosomes from MSCs have a beneficial impact on microglia, oligodendrocytes and neurons after injury across different disease models.

## 4. Pre-Clinical Data of Exosomes from MSCs in MS Models

As MS only occurs in humans, there is no animal model that addresses all aspects of the disease. The most commonly applied methods are murine models of CNS inflammation and demyelination. These include experimental autoimmune encephalomyelitis (EAE), cuprizone application, and infection with Theilers murine encephalomyelitis virus (TMEV). A number of studies have applied these, assessing the efficacy of exosomes and EVs from MSCs (summarized in Table 1).

Using an EAE rat model, the effect of exosomes from rat BM-MSCs was compared to that of BM-MSCs, and to rats receiving a vehicle as a control [64]. Treatments were administered intravenously 24 h after EAE induction. Two groups received exosomes, one with a dose of 100 μg and one with 400 μg exosomes. In both groups, levels of anti-inflammatory cytokines associated with the M2 phenotype of microglia were significantly upregulated, whereas pro-inflammatory cytokines associated with the M1 phenotype were downregulated. The exosome groups also displayed significantly increased protein and mRNA expression levels of M2 phenotype markers. Histologically, exosome treatment led to reduced infiltration of inflammatory cells in the CNS and decreased demyelination, with similar effects as injection of MSCs. The modulation of microglia from the M1 to M2 phenotype was confirmed by additional in vitro experiments. Overall, these findings suggest that exosomes recapitulate the beneficial effects of MSCs, displaying a shift in the microglial phenotype from inflammatory to anti-inflammatory, with reduced inflammation and demyelination leading to improved clinical outcomes.

Another study examined the effect of intravenously administered EVs from human adipose-derived MSCs in a demyelinating mouse model induced by TMEV [65]. A single dose of 25 μg of EVs or placebo was injected on day 60 postinfection and mice were sacrificed on day 75. Analysis of biodistribution showed EVs in the brain as well as peripheral organs, such as the lungs, spleen and liver. EVs resulted in decreased expression of the microglial marker iba-1 within brain tissue and changes in microglial morphology from rounded cell bodies with short processes to cells with longer, ramified processes, suggesting modulation of microglial activation from M1 to M2 phenotype. Concordantly, levels of inflammatory cytokines in plasma decreased in the EV-treated group. Immunofluorescence analysis showed that the myelin proteins CNPase and MPB increased in the EV-treated group, indicating remyelination. The reduced inflammation and increased myelin content led to improved motor behavior in mice receiving EVs, compared to sham treatment.

In a third pre-clinical study, EVs from MSCs were administrated intranasally in EAE mice [66]. The MSCs were derived from mouse adipose tissue. EVs at a dose of 10 μg were administered daily at the peak of disease, between days 15 and 27 post-immunization. Control groups received either PBS intranasally between day 15 and 27 or 500,000 MSCs intranasally at day 15 and 24. Both EV- and MSC groups exhibited immunomodulatory effects with an increase in regulatory T cells. Histological analysis with luxol fast blue showed reduced demyelination in mice receiving EVs, but not in mice receiving MSCs. However, clinical outcomes were improved in both treatment groups. These findings indicate that EVs may recapitulate the beneficial immunomodulatory and neuroprotective effects of MSCs.

A study from the US tested exosomes from MSCs in both EAE and cuprizone models [60]. Exosomes were harvested from MSCs derived from the BM of rhesus monkeys. In the EAE model, mice received 5 $\times \;10^{10}$ exosomes intravenously twice a week for 4 weeks, starting on day 10 following disease induction. In the cuprizone group, mice also received 5 $\times \;10^{10}$ exosomes intravenously once a week for two weeks initiated on the day of cuprizone diet withdrawal. Before this, mice had received cuprizone for 4 weeks. Labeling with green fluorescent protein (GFP) showed that the exosomes crossed the BBB and were localized within the parenchymal cells in the CNS four hours after administration. Exosomes led to improved remyelination in both models due to increased oligodendrocyte precursor cell (OPC) proliferation and differentiation. Also, microglia were modulated from the inflammatory M1 phenotype to the anti-inflammatory M2 phenotype, with decreased levels of IL-1β and TNFα and increased levels of IL-10 and TGFβ. The remyelinating and anti-inflammatory effects led to significantly improved neurological symptoms and cognitive function in the EAE and cuprizone models, respectively. The beneficial effects of the exosomes were mechanistically caused by the inactivation of the toll-like receptor 2 (TLR2) signaling pathway. Activation of TLR2 normally prevents remyelination via inhibition of OPC differentiation and also promotes neuroinflammation [67,68].

An Italian study assessed the therapeutic effect of small extracellular vesicles (sEVs) in a chronic EAE model [69]. Mice were treated with 5 μg sEVs from murine adipose-derived MSCs intravenously, either at day 3, 8 and 13 as a preventive treatment protocol, or at day 12, 16 and 20 as a therapeutic treatment protocol. Mice receiving the preventive treatment protocol experienced amelioration of disease with a reduction in symptoms, whereas this did not occur in the therapeutic protocol when the EAE already was established. In the preventive treatment protocol, sEVs led to inhibition of microglial activation, decreased T-cell extravasation and reduced demyelination, as compared to the mice in the control group receiving the carrier substrate.

A similar study used exosomes from MSCs obtained from the periodontal ligament of RRMS patients [70]. A total of 24 μg of exosomes was injected intravenously to the mice 14 days following EAE induction. Exosome treatment effectively reduced inflammation by blocking the NALP3 inflammasone and was associated with improved outcomes. TLR4 and nuclear factor (NF)-κB were elevated in control mice, but reduced in treated mice, suggesting that the effect of the exosomes occurred through inhibition of the TLR4 signaling pathway.

**Table 1 ijms-25-10292-t001:** Performed studies assessing stem-cell derived sEVs/exosomes in pre-clinical MS-models.

Author, Country, Year	Species	MS Model	Stem Cell Type	Stem Cell Source	EV/Exosome Dose	Exosome Numbers	Administration Mode	Time of Delivery after Disease Induction	Follow-Up	Effect
Li et al., China, 2019 [64]	Rats	EAE	BM-MSCs	Rats	100 μg, 400 μg	NS	IV	24 h	14 d	Modulation microglia M1–M2, Reduced demyelination, Improved outcomes
Laso-Garcia, Spain, 2018 [65]	Mice	TMEV	Adipose-derived MSCs	Human	25 μg	NS	IV	60 d	75 d	Reduced neuroinflammation, Reduced demyelination, Improved motoric function
Fathollahi, Iran, 2021 [66]	Mice	EAE	Adipose-derived MSCs	Mice	10 μg	NS	Intranasal	15–25 d	27 d	Reduced demyelination, Amelioration of clinical symptoms, Modulation of regulatory T-cells
Zhang, USA, 2022 [60]	Mice	Cuprizone and EAE	BM-MSCs	Rhesus monkey	$5\times 10^{10}$	$5\times 10^{10}$	IV	EAE: 10 d, Cuprizone: 5 weeks	EAE: 35 d, Cuprizone: 2 weeks	Exosomes within CNS after 4 h, Modulation microglia M1–M2, Increase OPC differentiation and myelination, Improved outcomes
Farinazzo, Italy, 2018 [69]	Mice	EAE	Adipose-derived MSCs	Mice	5 μg	NS	IV	Preventive: 3, 8 and 13 d, Therapeutic: 12, 16 and 20 d	36 d	In preventive protocol: Inhibition of microglial activation, Decreased T-cell activation, Decreased demyelination, No effect in therapeutic protocol
Rajan, Italy, 2017 [70]	Mice	EAE	Dental tissue	Human	24 μg	NS	IV	14 d	28 d	Reduced neuroinflammation, Inhibition TLR4 pathway, Improved outcomes
Jafarinia, Iran, 2020 [71]	Mice	EAE	Adipose-derived MSCs	Human	60 μg	NS	IV	10 d	30 d	Reduced neuroinflammation, Reduced demyelination, Improved outcomes
Koohsari, Iran, 2021 [72]	Mice	EAE	Umbilical cord	Human	50 μg	NS	IV	9 d	30 d	Reduced CNS leukocyte infiltration, Reduced inflammatory cytokines, Improved outcomes
Riazifar, USA, 2019 [73]	Mice	EAE	BM-MSCs	Human	150 μg	NS	IV	18 d	40 d	Decreased microglia infiltration, Less demyelination

EAE; experimental autoimmune encephalomyelitis, MSCs; mesenchymal stem cells, BM; bone marrow, NS; not specified, IV; intravenous.

EVs from human adipose-derived MSCs were assessed in another study [71]. EAE mice received either 60 μg exosomes, 1 $\times \;10^{6}$ MSCs, or carrier substrate intravenously on day 10 after disease induction. Both groups receiving exosomes and MSCs displayed reduced disease severity scores, with less inflammation and demyelination as compared to the untreated control group. Increased proliferation of regulatory T cells was hypothesized as a possible mechanism as mice receiving MSCs had significantly increased numbers in the spleen compared to control mice. However, this number was not significantly increased in mice receiving sEVs.

Another study also applied exosomes/EVs from the human umbilical cord in an EAE mouse model [72]. Mice received an intravenous injection of 50 μg exosomes on day 9 post-immunization. Exosome treatment led to improved outcomes with reduced CNS leukocyte infiltration along with decreased levels of the inflammatory cytokines IFN-γ, TNFα and IL-17a. This study also reported increased levels of regulatory T cells in the spleen, representing a possible causal factor for the beneficial treatment response. However, no differences were shown in the frequency of MBP-positive cells in the spinal cord, suggesting no impact on demyelination.

Exosomes from human BM-MSCs were used in a US study applying the EAE model [73]. Naïve and IFN-γ stimulated MSCs ($1\times 10^{6}$) and their respective exosomes (150 μg or $1\times 10^{9}$) were injected intravenously at the peak of the disease. In an additional group of mice, exosomes were labeled with lipophilic DiR dye. While the MSCs got trapped in the small lung vasculature, this did not occur with the exosomes. The IFN-γ stimulated exosome product was identified in the spinal cords of EAE mice at three hours following administration, but not in healthy animals. The signals disappeared after 24 h. Both exosomes from naïve MSCs and stimulated MSCs resulted in decreased microglia infiltration and less demyelination than in control mice. Inactivation of RNA in the exosomes reduced the treatment effect on regulatory T cells, underlining the important role of RNA in exosome functions. The morphology of the microglia changed from the typically activated swollen, amoeboid-like structure to long, ramified, resting microglia, the morphology which was observed in healthy mice. Both exosomes from IFN-γ stimulated and naïve MSCs significantly ameliorated the disease course, but the stimulated exosomes to a higher extent than naïve. Proteomic analysis displayed 104 unique peptides in the exosomes from the IFN-γ stimulated MSC, which were not present in the exosomes from normal MSCs. Several of these proteins, including macrophage inhibitory cytokine 1 (MIC-1), galectin-1 (Gal-1) and HSP70, are known to possess anti-inflammatory and neuroprotective properties. These results suggest that the therapeutic properties of the exosomes may be potentiated by modifying the MSCs prior to exosome harvesting.

In summary, pre-clinical data show the therapeutic potential of exosomes in rodent MS models. However, studies differ in administration mode, origin of exosomes, dose and timing of treatment application.

## 5. Exosomes from MSCs in MS—A Clinical Perspective

So far, no clinical trials have assessed the therapeutic effect of exosomes in MS. Different aspects are here relevant to consider, including safety matters, mode of administration and translatory issues.

### 5.1. Data on Safety

Sufficient data on safety are essential before considering a clinical trial with a new therapeutic. A number of small clinical trials have already been performed applying exosomes for other conditions than MS (Table 2).

Two studies and one case report have applied exosomes from MSCs intravenously. In a single-center, randomized, placebo-controlled clinical trial, exosomes from allogeneic MSCs derived from the umbilical cord were injected intravenously and intraarterially in patients with chronic kidney disease [74]. A total of 40 patients were included and randomized 1:1. Patients in the treatment group received 100 μg/kg exosomes per kg body weight weekly for two weeks. The first dose was given intravenously, the second dose intraarterially. Patients receiving saline as a placebo only received treatments intravenously. Results showed no significant adverse events related to treatment. Significant improvements were noted in glomerulation filtration rate and other biomarkers of kidney function during the 12-month study period. Exosome treatment was associated with a decrease in pro-inflammatory cytokines in serum. Kidney biopsies were performed in two of the treated patients and in one patient in the control group. According to the authors, there was more mRNA expression of CD133 and Ki67 in the treated patients, suggesting the growth of tubular epithelial cells after exosome treatment.

Another study assessed the effect of exosomes in patients with severe COVID-19. Here, 24 patients were treated with an intravenous dose of exosomes from allogeneic BM-MSCs [75]. Patients received 15 mL of ExoFlo©, a product developed by Direct Biologics™. The exact concentration/numbers of exosomes were not specified. No adverse events attributable to the exosome product were observed. The survival rate was 83% with 17/24 patients recovering and 3/24 patients remaining critically ill. Overall, significant improvements in clinical status, arterial oxygenation levels and lab status were noted. However, efficacy was hard to evaluate properly in the absence of a control group.

In a case report, increasing doses of exosomes were infused every 2–3 days in a patient with chronic graft-versus-host disease [76]. The EV fraction of the supernatant from $4\times 10^{7}$ MSCs (1.3–3.5 × $10^{10}$) that had been conditioned for 48 h was used and defined as one unit. MSCs were derived from four allogeneic BM donors. Doses were gradually increased until four units were reached at day 14. The next two weeks, symptoms improved dramatically along with a reduction in secreted cytokines. The patient died, however, 7 months later, due to pneumonia.

Other administration forms have also been explored. One study with healthy volunteers tested the inhalation of exosomes from allogeneic MSCs, without adverse events except for two episodes of asymptomatic bradycardia [77]. Exosomes from MSCs have also been injected locally into joints in patients with osteoarthritis [78] and applied as skin formulations for hyperpigmentation [79] and acne scars [80].

Recently, the results from a phase I/II trial assessing intranasally applied exosomes from allogenic MSCs in patients with Alzheimer´s disease were published [81]. A total of nine patients received exosomes in a dose-escalation design; three patients received 200 million exosomes per dose, three patients 400 million and three patients 800 million. Patients received two doses per week for 30 days and were followed for 48 weeks. There were no adverse events reported related to the study drug. The medium dose (400 million exosomes/dose) showed improved results in cognitive testing. Levels of amyloid and tau depositions were stable. The authors stated that the results could form the basis for a larger trial.

In conclusion, clinical experience so far has not identified adverse events related to exosome therapy. However, data are scarce, as few studies with few patients have been performed.

### 5.2. Mode of Administration

More than 98% of small molecule drugs are blocked by the BBB [82]. In clinical medicine, this barrier became apparent in 1914, as it was shown that salvarsan, a drug for syphilis, was unable to able to enter the brain for the treatment of neurosyphilis [83]. The brain parenchyma is safeguarded by the BBB, the CSF is regulated by the blood–CSF barrier. The latter barrier is mediated by the epithelial membranes of the choroid plexus, lining the floor of each of the four cerebral ventricles. Although both barriers are part of the CNS, they have different properties [84]. The BBB is formed by high resistance tight junctions between endothelial cells and has 300 times higher electrical resistance compared to the epithelial barrier of the choroid plexus [85,86]. Due to the low permeability of molecules and particles through the BBB, different routes of administration have been explored for drug delivery to the brain parenchyma. In 1982, trans-nasal administration of progesterone in monkeys was demonstrated to yield higher CSF levels than the intravenous route [87].

Regarding exosomes, studies so far have demonstrated good safety following both intravenous and intranasal administration. From a MS perspective, the intravenous application will unavoidably result in substantial dilution with few exosomes available to penetrate the BBB. Exosomes may be injected intrathecally, thereby circumventing the BBB. However, also here there is a problem with dilution due to the rapid turnover of CSF. Moreover, there is the risk of potentially serious side effects. An intranasal delivery mode allows for a minimally invasive and easily repeatable method and may provide a method of bypassing the BBB, thereby delivering the exosomes directly to the CNS. Due to their limited size, the exosomes may be taken up by the terminal axons of the trigeminal or olfactory nerve [88,89]. They may also penetrate the olfactory epithelium transcellulary or paracellulary.

In rodents, studies have demonstrated that the exosomes reside within the brain for 48 h after nasal administration, peaking between 3 and 12 h [90]. In a mouse model of traumatic brain injury, exosomes from human adipose-derived stem cells effectively infiltrated the brain following intranasal treatment [59]. The labeled exosomes colocalized with neurons, astrocytes and microglia, indicating presence within these cells. Results demonstrated significant recovery of motor deficits in mice receiving exosomes. No clinical studies have, however, so far assessed the rate of absorption of exosomes into the CNS via the nasal route.

### 5.3. Practical Issues of Exosomes as a Therapeutic Product

#### 5.3.1. Foreign Exosomes from Ex Vivo Expansion

Exosomes are secreted into the conditioned medium during the ex vivo expansion of MSCs. Protocols to expand MSCs often include supplements of fetal bovine serum (FBS) or human platelet lysate (hPL), both of which contain exosomes from their donors. In the production process, exosomes from other sources than MSCs are not wanted. Thus, serum- and xenogeneic-free media should ideally be used during the expansion of MSCs. Alternatively, exosomes may be collected during a starvation period of MSCs in basal medium without supplements and foreign exosomes. This may, however, affect the survival and properties of the MSCs and thus also affect the therapeutic effect of the exosomes. A third alternative is to apply ultracentrifugation or cross-flow filtration to deplete exosomes from FBS or hPL before use [91]. This may, however, increase the costs of the production process.

#### 5.3.2. Large Scale Production

The use of differential centrifugation has commonly been applied for exosome isolation in pre-clinical studies [92]. However, this method is labor-intensive and time-consuming, with relatively small capacity, limiting its potential for clinical use. Size exclusion chromatography is another potential method for pre-clinical use, but is unsuitable for large-scale production due to limited volumes. A recent study showed that the use of cross-flow filtration combined with ultracentrifugation resulted in sEVs with similar properties as compared to the current gold standard of differential centrifugation combined with ultracentrifugation [93]. In this study, cell culture in a hollow fiber bioreactor allowed for the large-scale expansion of MSCs and the production of several liters of conditioned medium. Also, the use of EV-depleted hPL did not alter MSC characteristics. This method, combining large-scale MSC expansion in a bioreactor and semi-automated sEV production, represents a promising way to produce exosomes for clinical trials under GMP conditions.

#### 5.3.3. Modification of MSCs

Because exosomes may be considered miniature versions of their parental cells, the contents and properties of exosomes depend on the state of the cell. The therapeutic potential of MSCs may be enhanced by pre-conditioning of the cells. Pre-conditioning of MSCs include physiological conditioning (hypoxic conditions and serum deprivation), molecular or pharmacological treatment (growth factors and spinal fluid), or thermal conditioning [94]. Recent studies have indicated improved neuroregenerative properties in vitro after such modifications to MSCs [95,96]. Likewise, anti-inflammatory and immunomodulatory properties of MSCs may be enhanced by pre-conditioning with pro-inflammatory cytokines [97,98]. An interesting question is whether exosomes from pre-conditioned MSCs possess increased therapeutic potential compared to exosomes of naïve MSCs. As previously mentioned, this was tested in a study of exosomes from MSCs stimulated with IFN-γ [73]. Here, clinical scores in EAE mice were significantly improved after intravenous administration of pre-conditioned MSCs as compared to naïve MSCs. Exosomes from IFN-γ stimulated MSCs also reduced macrophage/microglia infiltration more efficiently than naïve MSCs. This indicates that pre-conditioned MSCs produce exosomes with increased immunomodulatory potential. The same principle may be applied to the regenerative perspective. Exosomes from MSCs trans-differentiated in the oligodendroglial or neural direction may possess superior abilities for remyelination and neuronal regeneration, respectively. MSCs trans-differentiated into neuronal progenitor cells have already been tested in MS patients with promising results [99], although a recent placebo-controlled trial failed to show general improvement in neurological deficits [100]. In theory, exosomes may be “tailored” in the same manner for clinical use. However, trans-differentiation of MSCs may affect the secretory potential of the cells. If exosome secretion is reduced, more MSCs will be necessary to achieve sufficient volumes for treatment.

#### 5.3.4. Translation from Pre-Clinical Models

Recently, a meta-analysis was published assessing the efficacy of EVs from MSCs in pre-clinical MS models [101]. A total of 19 studies using different MS models were included. Results demonstrated that MSC-derived EVs significantly reduced disease severity (standardized mean difference 2.95%, VI 1.18–2.83, *p* < 0.0001), indicating a beneficial effect on demyelination, neuroinflammation and disease severity. The systematic review also highlighted significant heterogeneity in MSC origin, different doses, timing of treatment and applied administration forms. This variation may represent a problem for the translation of findings into clinical trials. In addition, a majority of studies only specified the volume and protein concentration of the solution with exosomes being tested. These quantities are, however, insufficient for dose calculation without knowing the concentration of exosome particles within the solution. Modern nanoparticle tracking equipment is able to quantify and characterize exosomes and thus provide an exact dose. Future pre-clinical and clinical studies should use this to shed information on the relationship between dose and effect, and to promote verifiability.

Because the production of clinical amounts of exosomes is expensive, the duration of a possible therapeutic effect is of large interest. Exosomes have been shown to impact target cells through epigenomic modification via miRNAs. This may imply that the therapeutic effect is more permanent than merely stimulation by growth factors. In this case, exosomes could be administrated less frequently, which would be relevant from a cost/benefit perspective. This aspect remains speculative, however, as data are lacking.

## 6. Conclusive Remarks

Although MS treatment has been revolutionized during the last decades for the benefit of MS patients, at least two important hurdles remain: (1) prevention of microglia activation driving disease progression and (2) promotion of remyelination and regeneration. Pre-clinical and clinical studies have shown a certain potential for MSCs to address both these issues. However, the administration of living cells is demanding from a logistic perspective and comes with the risk of cell-related complications. Also, there are little to no data concerning the fate of MSCs in MS patients. Likely, they only survive for a very limited time period. Because MSCs exert their therapeutic effects via their potent secretory function and not through cell replacement, an obvious alternative is, therefore, to apply their secreted products instead of the cells themselves. Exosomes represent an intriguing part of the MSC secretome and contain nucleic acids capable of the epigenetic modulation of target cells. Indeed, studies using rodent MS models have suggested that exosomes may be equally effective in preventing microglia activation and promoting remyelination as their parental cells [64,66].

Data suggest that these effects are mediated via the inhibition of TLR2 and TLR4 signaling pathways [60,70]. Despite the heterogeneity concerning the origin of the MSCs and the dose and administration mode, pre-clinical studies have consistently shown promising results of exosome treatment in different MS models. Clinical trials have already been performed for other conditions than MS, demonstrating the safety of exosome treatment without alarming adverse events. Recently, safety and feasibility were shown following intranasal application of exosomes from allogeneic MSCs in patients with Alzheimer’s disease. This approach could represent the next step in the translatory process of testing exosomes in MS patients.

Rodent studies have demonstrated that exosomes are absorbed within the CNS shortly after intranasal administration and taken up by neurons and glial cells, including microglia [59,90]. Given the current challenges within MS treatment today, a phase I/II trial of intranasally administrated exosomes testing the safety and efficacy on chronic microglial inflammation may be feasible. The intranasal method is not invasive, may be easily repeated and serious adverse events are unlikely, especially considering the acceptable safety profile demonstrated in the trial for Alzheimer’s disease. If feasible, such a study should also aim to assess the uptake of allogeneic exosomes into the CNS via the intranasal pathway.

In conclusion, exosomes from MSCs represent a promising treatment for MS-related microglial activation and for the promotion of neuroregeneration. A possible first step in the clinical translation may be to test intranasally administrated exosomes in MS patients targeting chronic, innate inflammation within the CNS. 

## Figures and Tables

**Figure 1 ijms-25-10292-f001:**
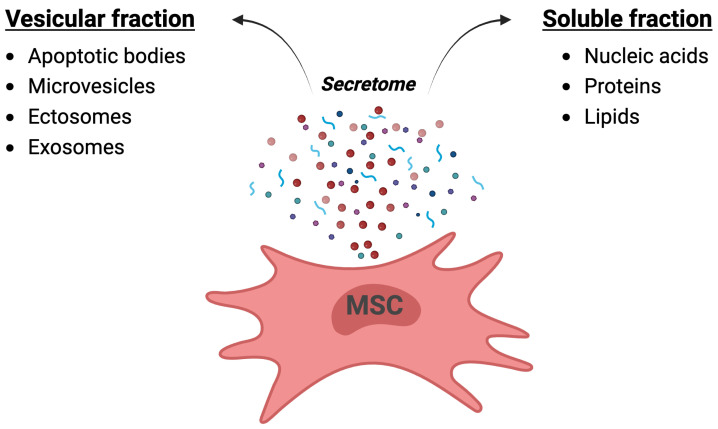
Illustration of the secretome released by mesenchymal stem cells. The soluble fraction (illustrated as blue strings) contains nucleic acids, proteins and lipids. The vesicular fraction (illustrated as round particles of different sizes) can contain apoptotic bodies, microvesicles, ectosomes and exosomes. Created with BioRender.com.

**Figure 2 ijms-25-10292-f002:**
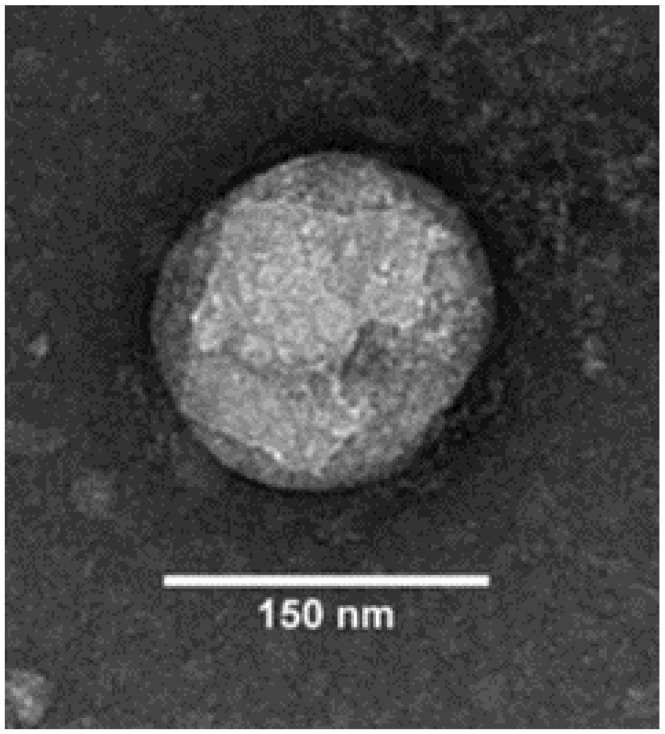
A single exosome from a mesenchymal stem cell imaged by transmission electron microscopy.

**Figure 3 ijms-25-10292-f003:**
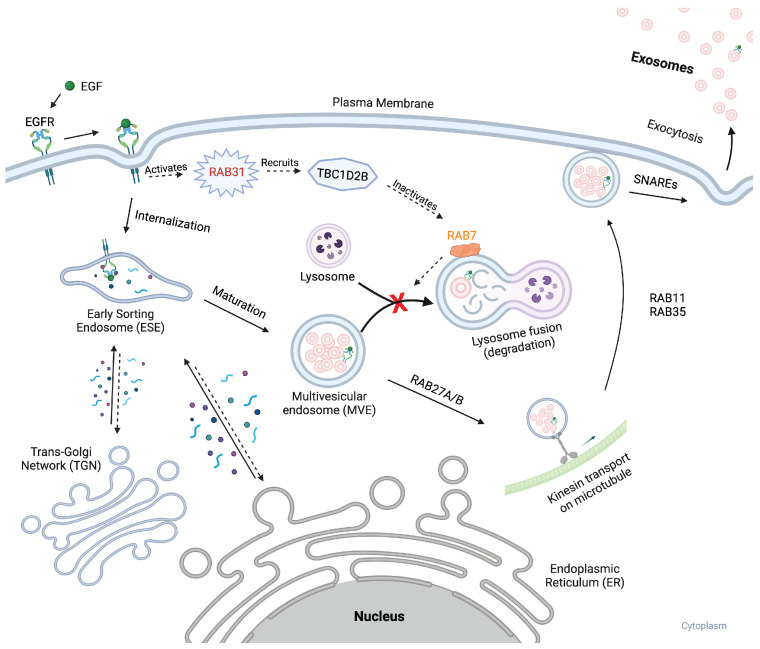
This figure illustrates the process by which multivesicular bodies (MVBs) can avoid lysosomal degradation and promote exocytosis. The pathway is initiated when epidermal growth factor (EGF) ligand binds to its receptor, epidermal growth factor receptor (EGFR) on the cell surface. This binding event triggers internalization and the inward budding of EGFR, activating RAB31. Activated RAB31 then recruits TBC1D2B, which inhibits RAB7, a key factor in MVB-lysosome fusion. By inhibiting RAB7, TBC1D2B prevents the fusion of MVBs with lysosomes, thus avoiding degradation. Consequently, RAB27A/B tethers the MVBs to kinesins, and transports them along the actin and microtubules toward the plasma membrane. The exosome then gets anchored to the membrane via. Rab GTPases such as RAB11 and RAB35. The fusion of MVBs with the plasma membrane, facilitated by SNARE proteins, allows for the release of exosomes into the extracellular space. Created with BioRender.com.

**Table 2 ijms-25-10292-t002:** Published studies assessing MSC derived sEVs/exosomes for different conditions.

Author, Country, Year	Condition	Administration Mode	N Patients	Study Type	Source Allogeneic MSCs	Dose	N Treatments	Follow-Up	Safety	Efficacy Compared to Control
Nassar, Egypt, 2016 [74]	Chronic kidney disease	IV/Intraarterial	40	Randomized, controlled	Umbilical cord	100 μg/kg bodyweight	2	12 months	No significant AEs	Improved GFR, Decreased pro-inflammatory cytokines
Sengupta, USA, 2020 [75]	COVID-19 and moderate/severe ARDS	IV	24	Single group	BM	NS (15 mL)	1	14 days	No AE attributable to IMP	NA (no control group)
Kordelas, Germany, 2014 [76]	GVHD	IV	1	Case report	BM	0.1–4 units (one unit = 1.3–3.5 × 10^10^ exosomes)	7	7 weeks	No AE attributable to IMP	NA (no control group)
Shi, China, 2021 [77]	Healthy volunteers	Inhalation	24	Single group	Adipose-derived MSCs	2 × 10^8^ particles to 16 × 10^8^ particles	1	7 days	Two asymptomatic bradycardia. No SAE	INA (no control group)
East, USA, 2020 [78]	Osteoarthritis	Injection	33	Single group	BM	60 × 10^11^ exosomes	1	6 months	No AE attributable to IMP	NA (no control group)
Cho, South Korea, 2020 [79]	Hyperpigmentation	Ointment	24	Randomized, controlled	Adipose-derived MSCs	2 × 10^10^ particles/mL	Twice per day for 8 weeks	8 weeks	No AE attributable to IMP	Reduced amounts of melanin
Kwon, South Korea, 2020 [80]	Skin scars	Gel	25	Single group, split face	Adipose-derived MSCs	1.6–9.8 × 10^10^ exosomes/mL	4	6 weeks	No AE attributable to IMP	Milder erythema on the exosome-treated side, Atrophic scar volume was significantly decreased
Xie, China, 2023 [81]	Alzheimer	Intranasal	9	Dose escalation	Adipose-derived MSCs	2–8 × 10^8^ exosomes	Twice/week for 12 weeks	48 weeks	No AE attributable to IMP	NA (no control group)

MSCs; mesenchymal stem cells, BM; bone marrow, NA; not applicable, IV; intravenous.

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
