# Peer review of "The Therapeutic Potential of Exosomes from Mesenchymal Stem Cells in Multiple Sclerosis"

_ijms, 2024, doi:10.3390/ijms251910292_

Round 1

Reviewer 1 Report

Comments and Suggestions for Authors

In this review paper, the authors discuss the role of mesenchymal stem cell (MSC)-derived exosomes in the therapy of multiple sclerosis (MS). The paper is well-written and provides a detailed discussion on the formation of exosomes. However, it appears incomplete. The authors seem to aim at highlighting the significance of MSC-derived exosomes in treating MS, particularly through the modulation of microglial functions. To achieve this, several additions are necessary:

-       Classification of MS: The paper should include a classification of MS, as the types of MS differ in their pathology. This section should address the role of microglial activation in each MS type and discuss how microglia influence other cell types involved in MS pathology. A figure illustrating these aspects would enhance clarity.

-       MSC Secretome: The review focuses on exosomes derived from MSCs but briefly mentions the MSC secretome. It would be beneficial to list MSC-derived growth factors and cytokines, alongside the content of MSC-derived exosomes. A subsection dedicated to the miRNA content of MSC-derived exosomes and their regulatory effects on microglia should be included. Based on the cytokine, growth factor, and miRNA content, the authors should explore both known and potential signaling pathways that MSC-derived exosomes might modulate in MS. Merely stating that "MSC exosomes regulate microglia via pathways such as the TLR4/NF-κB/PI3K/AKT cascade, influenced by miR-216a-5p" is not so attractive.

-       Blood-Brain Barrier (BBB): The paper should include a section on the BBB. Although it is mentioned that exosomes might pass through the BBB, there are instances where the BBB does not allow particles smaller than exosomes. Also, inflammatory conditions might make MSCs a preferable option as they can target lesion sites via chemokines. Additionally, maintaining the integrity of exosomes in the bloodstream and avoiding off-target delivery could be challenging. These points need some discussion. This section should also discuss the BBB's condition in different MS types, how microglia affect the BBB, potential modifications to exosomes to facilitate BBB passage, and the impact of MSC-derived exosomes on the BBB. A figure illustrating the role of the BBB in MS, the role of microglia in BBB maintenance, BBB disruption in MS, and how exosomes affect microglia-mediated BBB damage would be valuable.

-       Figure 3: This figure requires modification. Clarify whether RAB31 and TBC1D2B are recruited to internalized vesicles or if RAB31 activates and inhibits RAB7, thereby affecting lysosomal fusion. The current representation suggests that RAB31 independently inhibits RAB7 rather than on the MBV surface. Similarly, clarify whether RAB27 is recruited to the MBV surface. To reduce confusion, consider depicting two pathways: one showing endocytosis and another illustrating exosome formation and the role of RAB31 in protecting against lysosome-mediated exosome degradation.

These revisions will help provide a more comprehensive and clear understanding of the role of MSC-derived exosomes in MS therapy.

Author Response

Reviewer: “Classification of MS: The paper should include a classification of MS, as the types of MS differ in their pathology. This section should address the role of microglial activation in each MS type and discuss how microglia influence other cell types involved in MS pathology. A figure illustrating these aspects would enhance clarity.” 

Reply: Thank you for your comment. We have added this to the start of the introduction: 

The core MS phenotypes are those of relapsing-remitting disease (RRMS) dominated by inflammation mediated by the adaptive immune system, and progressive disease, dominated by chronic microglia-driven inflammation and neuronal degeneration. RRMS is characterized by the invasion of macrophages, microglia and lymphocytes through the injured blood-brain barrier (BBB) into the CNS parenchyma with the formation of inflammatory and demyelinating lesions in both white and grey matter. Microglial activation has been found in the white matter remote from the active lesions, where they may represent the earliest stage of lesion development \cite{vanderValk2009}. Microglia also play a role in clearance of myelin debris, which is necessary for remyelination to occur. The remyelination in patients is, however, incomplete with thinner and more fragile myelin sheaths than normal myelin \cite{Franklin2022}.   

After typically 15-20 years, patients may experience a progressive clinical course termed secondary progressive MS (SPMS), whereas approximately 15\% of patients have a progressive course from the beginning, termed primary progressive MS (PPMS). Neurodegeneration is the hallmark of progressive MS disease, which typically leads to immobility and cognitive decline. Progressive MS is also characterized by a compartmentalized chronic inflammation with focal and diffuse microglial activation behind an intact BBB. 

Reviewer: “MSC Secretome: The review focuses on exosomes derived from MSCs but briefly mentions the MSC secretome. It would be beneficial to list MSC-derived growth factors and cytokines, alongside the content of MSC-derived exosomes. A subsection dedicated to the miRNA content of MSC-derived exosomes and their regulatory effects on microglia should be included. Based on the cytokine, growth factor, and miRNA content, the authors should explore both known and potential signaling pathways that MSC-derived exosomes might modulate in MS. Merely stating that "MSC exosomes regulate microglia via pathways such as the TLR4/NF-κB/PI3K/AKT cascade, influenced by miR-216a-5p" is not so attractive.” 

Reply: Concerning the secretome, we have now implemented a section about the content of the secretome (section 2): 

MSCs are highly secretory, and their secretome contains a soluble and a vesicular fraction (Figure 2). The soluble fraction contains a plethora of neurotrophic growth factors, chemokines and cytokines such as IL-6, IL-10, IL-17, PGE2, CXCL-10, glial-derived neurotrophic factor (GDNF), brain-derived neurotrophic factor (BDNF), vascular endothelial growth factor (VEGF), fibroblast growth factor (FGF), hepatocyte growth factor (HGF), nerve growth factor (NGF) and insulin-like growth factors 1 and 2 (IGF-1 and IGF-2) \cite{Pinho2020} \cite{L2019}. The vesicular fraction contains extracellular vesicles (EVs) of different sizes, including exosomes. 

Concerning miRNAs, we have now edited this section (section 3.3):  

Moreover, miR-216-5p \cite{Li2020}, miR-146a-5p \cite{Zhang2021} and miR-125a \cite{Chang2021} have also been demonstrated to alleviate pro-inflammatory microglia following CNS injury. Exosomes from MSCs modulate microglia from inflammatory to anti-inflammatory phenotype by inhibition of the TLR4/NF-κB/PI3K/AKT inflammatory cascade \cite{Liu2020}. This was shown in a rodent model where exosomes were administrated intravenously as a single injection after SCI. This led to significantly improved outcomes in mice receiving exosomes as compared to controls. Interestingly, exposure of MSCs to hypoxia improved the anti-inflammatory effect of the resulting exosomes. 

Reviewer: “Blood-Brain Barrier (BBB): The paper should include a section on the BBB. Although it is mentioned that exosomes might pass through the BBB, there are instances where the BBB does not allow particles smaller than exosomes. Also, inflammatory conditions might make MSCs a preferable option as they can target lesion sites via chemokines. Additionally, maintaining the integrity of exosomes in the bloodstream and avoiding off-target delivery could be challenging. These points need some discussion. This section should also discuss the BBB's condition in different MS types, how microglia affect the BBB, potential modifications to exosomes to facilitate BBB passage, and the impact of MSC-derived exosomes on the BBB. A figure illustrating the role of the BBB in MS, the role of microglia in BBB maintenance, BBB disruption in MS, and how exosomes affect microglia-mediated BBB damage would be valuable.” 

Reply: We appreciate thorough feedback. We have now included the following paragraph in section 5.2 of our revised manuscript: 

More than 98% of small molecule drugs are blocked by the BBB \cite{Pardridge2005}. In clinical medicine, this barrier became apparent in 1914, as it was shown that salvarsan, a drug for syphilis, was unable to able to enter the brain for treatment of neurosyphilis \cite{Mcintosh1914}. Whereas the brain parenchyma is safeguarded by the BBB, the CSF is regulated by the blood-CSF barrier. The latter barrier is mediated by the epithelial membranes of the choroid plexus, lining the floor of each of the four cerebral ventricles. Although both barriers are part of the CNS, they have different properties \cite{Brightman1969}. The BBB is formed by high resistance tight junctions between endothelial cells and has 300 times higher electrical resistance compared to the epithelial barrier of the choroid plexus \cite{Zeuthen1981} \cite{Smith1986}. Due to the low permeability of molecules and particles through the BBB, different routes of administration have been explored for drug delivery to the brain parenchyma. In 1982, trans-nasal administration of progesterone in monkeys was demonstrated to yield higher CSF levels than the intravenous route \cite{AnandKumar1982}. 

Reviewer: “Figure 3: This figure requires modification. Clarify whether RAB31 and TBC1D2B are recruited to internalized vesicles or if RAB31 activates and inhibits RAB7, thereby affecting lysosomal fusion. The current representation suggests that RAB31 independently inhibits RAB7 rather than on the MBV surface. Similarly, clarify whether RAB27 is recruited to the MBV surface. To reduce confusion, consider depicting two pathways: one showing endocytosis and another illustrating exosome formation and the role of RAB31 in protecting against lysosome-mediated exosome degradation. 

These revisions will help provide a more comprehensive and clear understanding of the role of MSC-derived exosomes in MS therapy.” 

Reply: We have revised Figure 3 to clarify the pathways involved by adding descriptive text to the arrows in the Rab31 pathway. Rab7 has been placed on the MVE surface. We also merged the lysosome and MVE arrows, and clearly pointed the Rab7 protein towards the merging point (X) to signify its role in inhibiting lysosome-MVE fusion. Upon EGF binding to EGFR, Rab31 is activated, initiating the internalization process. Concurrently, Rab31 recruits TBC1D2B, which inactivates Rab7, thus preventing the lysosome-MVE fusion and therefore promoting exosome secretion. Both the internalization process and the Rab31 mediated pathway occur in parallel, which the new modifications made to the figure, should now clarify. 

Reviewer 2 Report

Comments and Suggestions for Authors

The authors have conducted a review on the therapeutic potential of mesenchymal stem cell-derived exosomes for multiple sclerosis. This review sequentially explains the pathophysiology of multiple sclerosis, mesenchymal stem cell-derived exosomes, their effects on the central nervous system, and data from animal models of multiple sclerosis, which helps enhance the reader’s understanding. Furthermore, it comprehensively discusses the prospects for expansion into clinical applications for multiple sclerosis patients, making it a meaningful contribution for publication as a review.

Author Response

We thank the reviewer for the feedback.

Reviewer 3 Report

Comments and Suggestions for Authors

In the present study, authors described the therapeutic potential of exosomes derived from mesenchymal stem cells (MSCs) targeting multiple sclerosis. In general, this article is well organized and well written. However, it is known that various cell types other than MSCs such as neural stem cells (Stem Cell Res Ther. 2023; 14: 198; Front. Cell Dev. Biol., 13 May 2021), endothelial cells (Scientific Reports volume 14, 4465, 2024; Front. Immunol., 28 July 2023), and astrocytes (Scientific Reports volume 14, 5272, 2024) produce exosomes as well. So, please discuss which is similar and which is different among on cell types regarding the traits of exosomes.

And also, although authors described the relationships between MSC exosomes and microglia (3.3 section), brain is composed of various cell types, such as neurons, astrocytes, oligodendrocytes, endothelial cells, and pericytes other than microglia. Therefore, it would be better that authors addressed the relationships between MSC exosomes and these cell types as well.

Similar concept paper is recently published by other groups (Molecular and Cellular Biochemistry (2024) 479:1643–1671). Please refer to it and clarify the point which is new compared with this article.

Author Response

Reviewer: “In the present study, authors described the therapeutic potential of exosomes derived from mesenchymal stem cells (MSCs) targeting multiple sclerosis. In general, this article is well organized and well written. However, it is known that various cell types other than MSCs such as neural stem cells (Stem Cell Res Ther. 2023; 14: 198; Front. Cell Dev. Biol., 13 May 2021), endothelial cells (Scientific Reports volume 14, 4465, 2024; Front. Immunol., 28 July 2023), and astrocytes (Scientific Reports volume 14, 5272, 2024) produce exosomes as well. So, please discuss which is similar and which is different among on cell types regarding the traits of exosomes.” 

Reply: We thank the reviewer for the feedback. We have now specified that virtually all cells in the human body produce and secrete exosomes, including neural stem cells, endothelial cells and astrocytes (section 3): 

Exosomes are a subpopulation of EVs produced and secreted by virtually all cells. 

However, this review focuses on exosomes from MSCs in a therapeutic context. Consequently, we think it would be out of scope to discuss exosomes from other cell types in detail in this review. 

Reviewer: “And also, although authors described the relationships between MSC exosomes and microglia (3.3 section), brain is composed of various cell types, such as neurons, astrocytes, oligodendrocytes, endothelial cells, and pericytes other than microglia. Therefore, it would be better that authors addressed the relationships between MSC exosomes and these cell types as well.“ 

Reply: We thank the reviewer for the thorough comments and have now implemented this into section 3.3 of the revised manuscript:

Exosomes applied for therapeutic purposes also impact oligodendrocytes and neurons within the CNS. Previous studies have demonstrated that exosomes from stem cells are colocalized with neurons and oligodendrocytes following intranasal administration, suggesting that the exosomes are absorbed by these cells \cite{Moss2021}. Another study applying the cuprizone and EAE models in mice, demonstrated increased number of newly generated oligodendrocytes in addition to mature oligodendrocytes with higher levels of myelin basic protein following intravenous administration of exosomes from MSCs \cite{Zhang2022}.  
Likewise, an in vitro model of ischemic stroke showed a beneficial effect of MSC exosomes as miR-134 promoted survival in oligodendrocytes by negatively regulating the caspase 8-dependent apoptosis pathway. There is also some evidence that exosomes may affect neurons. In a rodent model of Alzheimer’s disease, MSC-derived exosomes inserted via stereotaxic surgery resulted in increased number of neural precursor cells with the subventricular zone of the brain, suggesting that exosomes accelerated neurogenesis \cite{CanalesAguirre2019}. Findings were associated with improved cognitive results in mice receiving exosomes. Exosomes have also been shown to promote neurogenesis and improve outcomes in models of ischemia \cite{Yang2017} and traumatic brain injury \cite{Zhang2017}.  

Taken together, evidence from pre-clinical studies indicate that exosomes from MSCs have a beneficial impact on microglia, oligodendrocyte and neurons after injury across different disease models. 

Reviewer: “Similar concept paper is recently published by other groups (Molecular and Cellular Biochemistry (2024) 479:1643–1671). Please refer to it and clarify the point which is new compared with this article.” 

Reply: The mentioned paper seems to have a more general focus on MSCs, as it summarizes all clinical studies that have been performed assessing MSCs in patients with MS. Our review focus more directly on exosomes from MSCs. In addition, our review summarize all clinical studies that have assessed exosomes and all rodent MS studies that have been published so far.